# KEYS TO ROBUST EDITS: FROM THEORETICAL INSIGHTS TO PRACTICAL ADVANCES

## ABSTRACT

Large language models (LLMs) have revolutionized knowledge storage and retrieval, but face challenges with conflicting and outdated information. Knowledge editing techniques have been proposed to address these issues, yet they struggle with robustness tests involving long contexts, paraphrased subjects, and continuous edits. This work investigates the cause of these failures in locate-and-edit methods, offering theoretical insights into their key-value modeling and deriving mathematical bounds for robust and specific edits, leading to a novel 'group discussion' conceptual model for locate-and-edit methods. Empirical analysis reveals that keys used by current methods fail to meet robustness and specificity requirements. To address this, we propose a Robust Edit Pathway (REP) that disentangles editing keys from LLMs' inner representations. Evaluations on LLaMA2-7B and Mistral-7B using the CounterFact dataset show that REP significantly improves robustness across various metrics, both in-domain and out-of-domain, with minimal trade-offs in success rate and locality. Our findings advance the development of reliable and flexible knowledge updating in LLMs.

## 1 INTRODUCTION

Large language models (LLMs, OpenAI 2023; Touvron et al. 2023a;b) have revolutionized the storage and retrieval of vast amounts of human knowledge. However, their training on extensive internet-based datasets inevitably leads to the incorporation of conflicting and rapidly outdated information, potentially contributing to the phenomenon of model hallucination (Zhang et al., 2023). To mitigate such issues, researchers have proposed knowledge editing techniques (Sinitsin et al., 2020; De Cao et al., 2021), which employ carefully designed, constrained updates to specific parameters. While giving promising results (Mitchell et al., 2022a; Meng et al., 2023a), there still exist challenges in its implementation, particularly their susceptibility to failure with robustness tests, e.g., when faced with long contexts, paraphrased subjects, and continuous edits (Ma et al., 2024c; Yang et al., 2024). For example, in Figure 1, although one can edit the knowledge 'Slovenia belongs to the continent of' from 'Europe' to 'Antarctica', the model still falls into failure when the subject is rephrased to 'The Republic of Slovenia'.

We aim to investigate the cause of such editing failures and further resolve them, with a specific focus on the widely recognized *locate-and-edit* methods (Meng et al., 2023a; Li et al., 2024a; Hu et al., 2024b; Meng et al., 2023b), which use causal interventions to pinpoint factual knowledge in the early-middle MLP layers of LLMs, treating these modules as associative key-value memories (Kohonen, 1972) and constructing new memories for editing. We give a new theoretical perspective into key-value modeling, demonstrating that any value retrieved from the module can be viewed as a linear combination of existing memories, where the weights are determined by similarities of the keys. As a result, knowledge insertion can be viewed as a localized patch rather than a comprehensive rewrite. We further provide a novel conceptual model, 'group discussion'. In this model, we view knowledge editing as inserting influence of a new voice among a mixture of opinion holders, and effective knowledge insertion requires a strong voice (i.e., large value) at the correct position (i.e., similar keys) while balancing the risk of affecting nearby group (i.e., related knowledge).

In light of the group discussion view, we empirically analyze the keys used by locate-and-edit methods and find they cannot follow the ideal scenario. In particular, we find that keys of robustness queries such as the same subject (e.g., Slovenia) with different surface forms (e.g., The Republic of Slovenia),

Figure 1: An example of the edited knowledge 'Slovenia belongs to the continent of' through knowledge editing and its failures on the different scenarios.

shuffled token ordering (e.g., ia Sloven) and appending long contexts (e.g., Tiffany ... Slovenia), are highly dissimilar to the edit key in the representation space. In contrast, keys from different subjects (e.g., Slovenia vs Croatia) could be highly similar, when subjects demonstrate semantic similarities. These make it impossible for the edited knowledge to both affect robustness queries while not affecting those semantically similar but different subjects. To mitigate such problems, we propose to build a Robust Edit Pathway (REP) which disentangles the keys used by editing methods from the inner representations of LLMs. Specifically, we train an adaptor that applies a gate mechanism to separate the keys from the model's inner representations, aggregating the keys for the parts that need editing, while keeping other representations intact. This adaptor is designed by aggregating semantically identical keys, and allows robust edit.

We evaluate REP on widely used LLaMA2-7B (Touvron et al., 2023b) and Mistral-7B (Jiang et al., 2023) using the CounterFact dataset, comparing it against an representative method ROME (Meng et al., 2023a). The Robust Edit Pathway leads to substantial improvements in performance, across various robustness metrics, both for in-domain and out-of-domain tests, with only a minimal trade-off in success rate and locality. These results not only validate our theoretical and empirical findings, but also demonstrate the adaptor's ability to generalize across different types of perturbations, effectively addressing the critical robustness issues in existing knowledge editing techniques and moving closer to our goal of reliable and flexible knowledge updating in large language models.

## 2 RELATED WORK

**Knowledge Editing.** As large language models have grown in complexity and size, post-modification has become increasingly challenging due to their opaque mechanisms and vast parameter spaces (Mitchell et al., 2022c; Zhong et al., 2023). This has led to heightened interest in knowledge editing, a technique for precise model modification. Knowledge editing are applied to various scenarios, such as editing for safety (Wang et al., 2024c), debias (Yan et al., 2024) and concepts (Wang et al., 2024d).

Knowledge editing approaches can be broadly categorized into two groups: those that preserve model parameters and those that modify them (Yao et al., 2023). Parameter-preserving methods include memory-based strategies (Mitchell et al., 2022d; Zhong et al., 2023; Hartvigsen et al., 2023) and alternative approaches (Dong et al., 2022; Huang et al., 2023) incorporate additional parameters into the model. On a different line of research, some work focuses on directly updating model parameters to avoid cost of an external memory, which can be further divided into meta-learning methods (Cao et al., 2021; Mitchell et al., 2022b; Tan et al., 2024) and locate-and-edit methods.

Our work is in line with the *locate-then-edit* methods, which draw much attention as they potentially unveil how the knowledge are stored in an LLM. These approaches first identify relevant parameters before updating them to modify specific knowledge, including KnowledgeNeuron's attribution-based neuron updating (Dai et al., 2022), ROME's causal mediation analysis for MLP editing (Meng et al., 2023a), MEMIT's multi-layer residual distribution (Meng et al., 2023b), PMET's refined allocation

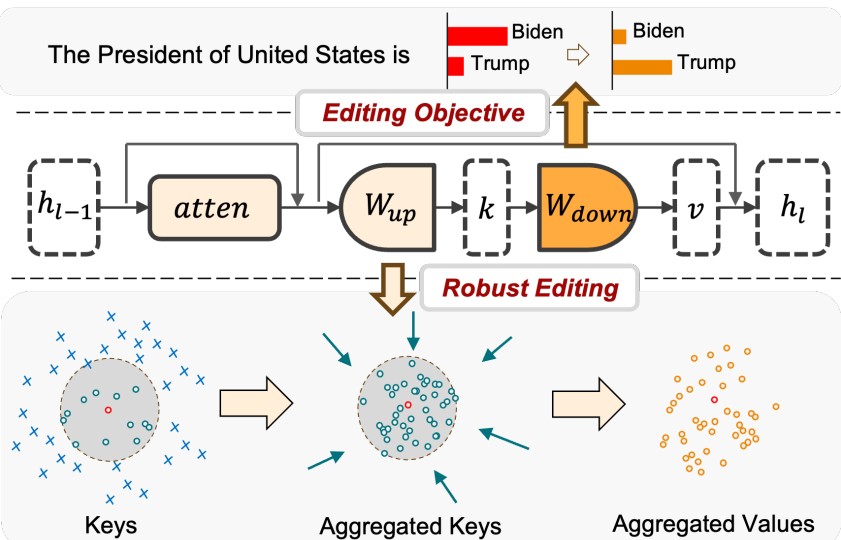

Figure 2: **Top**: Illustration of the knowledge editing task. **Middle**: For a specific layer, the *locate-and-edit* methods use representations inside FFN module as keys and values. **Bottom**: The keys are diverse and dissimilar, our REP aggregates the keys to facilitate robust editing.

strategy (Li et al., 2024a), and WilKE's dynamic layer selection (Hu et al., 2024b) to reduce potential negative effects. These methods all utlize inner representations as keys for key-value modeling. In contrast, we show that inner representations cannot meet the requirements of robust and specific edits, and we propose branching path adaoptor to mitigate this.

**Issues of Knowledge Editing.** Despite the promise of knowledge editing for refining large language models, various challenges persist in practical applications. Edits often degrade general language abilities (Gu et al., 2024; Ma et al., 2024b), damage the hidden space (Wang et al., 2024b), struggle to propagate to related facts (Hua et al., 2024), and are easily forgotten during sequential updates (Gupta et al., 2024). Moreover, multi-hop reasoning can elicit old knowledge (Zhang et al., 2024), and models may collapse after few edits (Yang et al., 2024; Brown et al., 2023).

Further complications include cross-lingual inconsistencies (Wang et al., 2024a), knowledge conflicts (Li et al., 2024b), and inadequate evaluation in realistic settings such as long-form generation (Rosati et al., 2024) and neighborhood knowledge (Ma et al., 2024a). These issues underscore the need for more sophisticated and comprehensive editing techniques. However, previous research largely remain focused on the outcomes of knowledge editing in various scenarios, lacking a deeper understanding of the underlying mechanisms of these algorithms and the true reasons behind their frequent failures. Our work presents both theoretical and empirical understanding regarding the reason of robustness failures of locate-and-edit methods, and proposes REP to enhance them.

## 3 KNOWLEDGE EDITING

In this section, we first formulate the task of knowledge editing and review the *locate-then-edit* methods.

**Task Definition** Given a knowledge triple: $(h, r, t)$, e.g., (USA, president of, Biden), knowledge editing aims to update the original target of the triple, i.e., 'Biden', to a new target $t_*$, say 'Trump'. Recent knowledge editing methods formulate such a task with a conditional generation. Following Meng et al. (2023a) and Meng et al. (2023b), we define a knowledge $\mathbf{f}$ as a triple $(h, r, t)$ where $h$ denotes the head entity, $r$ the relation, and $t$ the tail entity (e.g., ($h$ =USA, $r$ =president of, $t$ =Biden)).

**Definition 3.1** (**Task Definition of Knowledge Editing**). *Given a knowledge triple $\mathbf{f} = (h, r, t)$, knowledge editing algorithm $\mathcal{A}$ aims to update the original knowledge in the language model $\mathcal{M}$ to*

*new knowledge* $\mathbf{f}' = (h, r, t_*)$. *This task can be formally expressed as follows:*

$$\mathcal{M}' = \mathcal{A}(\mathcal{M}), \quad such\ that \quad k' \in \mathcal{M}', k \notin \mathcal{M}',$$

Autoregressive transformer-based large language models (LLMs) can answer natural-language queries by leveraging implicit knowledge encoded within their parameters. For instance, given the prompt "The president of USA is", a well-trained model would likely respond with "Biden".

Building on this, the *Locate-then-Edit* approach, specifically in the context of ROME, demonstrates that modifying specific MLP layers locating in the critical path of an LLM suffices to edit its factual associations. This method views the MLP layers $W$ in the transformer as a form of linear associative memory, observing that any linear operation $W$ can function as a key-value store, where a set of vector keys $K = [k_1|k_2|...]$ corresponds to a set of vector values $V = [v_1|v_2|...]$.

**Definition 3.2** (**The Solution of ROME**). *In ROME, a new key-value pair* $(k_*, v_*)$ *can be inserted into the language model using the following closed-form solution:*

$$minimize\ ||\hat{W}K - V||\ such\ that\ \hat{W}k_* = v_*,\ by\ setting\ \hat{W} = W + \mathbf{\Lambda}(C^{-1}k_*)^T$$

*where:*

- $C = KK^T$ *is a constant matrix pre-cached by estimating the uncentered covariance of $k$ from a sample of Wikipedia text,*
- $\mathbf{\Lambda} = \frac{v_* - Wk_*}{(C^{-1}k_*)^T k_*}$ *is a vector proportional to the residual error of the new key-value pair on the original memory matrix.*

To implement this solution, it is necessary to extract the key $k_*$ and calculate the value $v_*$.

**Remark 3.3** (Extract $k_*$). *In* $\mathcal{M}$, $k_*$ *is obtained by averaging the activations collected at the last token of the head entity $h$, processing a small set of texts that end with the head entity $h$. This can be formally written as:*

$$k_* = \frac{1}{M} \sum_{i=1}^{M} k(x_j + h),$$

*where $k(\cdot)$ is the input of the second MLP layer of the $l_*$-th FFN layer in the transformer, $M$ is the number of the selected texts and $x_j$ represents a random prefix.*

Once $k_*$ is extracted, the next step is to determine the appropriate value $v_*$ for the new key-value pair.

**Remark 3.4** (Calculate $v_*$). *Let* $\mathbb{P}_{\mathcal{M}'}(t_*|p)$ *denote the probability of $t_*$ after $\mathcal{M}$ processes query prompt $p$. We seek a vector $\mathbf{z}$ to substitute as the output of the MLP in layer $l^*$ at token $i$ (denoted $m_i^{(l*)} : \mathbf{z}$) such that the network predicts the target tail entity $t_*$ while maintaining the model's understanding of the subject's essence. The optimization objective is as follows:*

$$v_* = \arg\min_{\mathbf{z}} \frac{1}{N} \sum_{j=1}^{N} \underbrace{- \log \mathbb{P}_{\mathcal{M}(m_i^{(l*)}:\mathbf{z})}[h'|x_j + p]}_{(a)\ Maximizing\ h'\ probability} + \underbrace{D_{KL}\left( \mathbb{P}_{\mathcal{M}(m_i^{(l*)}:\mathbf{z})}[\cdot|p'] || \mathbb{P}_{\mathcal{M}}[\cdot|p'] \right)}_{(b)\ Controlling\ essence\ drift}.$$

*where $p'$ is 'subject is a'.*

In conclusion, the ROME method effectively enables the insertion of new knowledge triples $(h, r, t_*)$ into large language models through operating key-value pairs.

## 4 THEORETICAL RESULTS OF KEY-VALUE ASSOCIATIVE MEMORY

The idea of keys and values in associative memory is analogous to the key-value databases in modern computer systems. What makes the difference here is that our FFNs implement a fuzzy retrieval mechanism, whereas the modern key-value databases generally require the keys to be unique.

Given a set of keys $K = [k_1|k_2|\cdots]$ and values $V = [v_1|v_2|\cdots]$, recall that locate-and-edit methods such as ROME, MEMIT, and PMET view the down project feed-forward $W$ as a linear associative memory, where it satisfies $WK \approx V$.

**Lemma 4.1** (**Fuzzy Key-Value Mapping**). *Given $K \in \mathbb{R}^{D_1 \times N}$ and $V \in \mathbb{R}^{D_2 \times N}$ that are already stored in the feed-forward layer $W \in \mathbb{R}^{D_2 \times D_1}$, assume $N \gg D_1$ and $K$ has the rank of $D_1$. When a new query $\hat{k}$ comes, its corresponding value can be represented as the weighted sum of existing values, $\hat{v} = \sum_i^N \alpha_i v_i$ and $\alpha = K^T (KK^T)^{-1} \hat{k}$ can be solved by the Moore-Penrose pseudoinverse.*

The above lemma demonstrates that the retrieved memory of a new test query can be considered as the linear combination of previously stored memory, which leads to two direct corollaries.

**Corollary 4.2** (**Edited Key-Value as a Patch against Original Knowledge**). *Recall that in locate-and-edit algorithms, a newly edited knowledge is injected into the memory as a new key-value pair $(k_*, v_*)$. Suppose the injection is lossless[1] and we have a set of key-value pairs that represent the same semantic meaning $k_i \in K_s, v_i \in V_s$ that are already stored in the memory. Then, if $K_s$ has full row rank, a query of $k \in K_s$ would retrieve a value $v = \sum_i \alpha_i v_i + \alpha_* v_*$.*

**Remark 4.3.** *This corollary demonstrates that the editing process does not essentially 'change' the previously stored knowledge. Instead, it leaves the previously stored knowledge intact and counters them with a newly added value $v_*$.*

**Lemma 4.4** (**Bound on optimized $\Delta v = v_* - v_o$**). *Assume the edited layer is only connected to the final prediction layer via an attention layer, where the attention layer has parameters $W_Q, W_K, W_V$, and $w_t$ and $w_{t_*}$ are the output embeddings for the original and edited target, we have the following inequality,*

$$(w_{t_*} - w_t)^T W_V (v_* - v_o) > \epsilon_1 + \epsilon_2 \tag{1}$$

$$\text{Cauchy-Schwarz} \Rightarrow ||(w_{t_*} - w_t)^T W_V|| \cdot ||v_* - v_o|| > \epsilon_1 + \epsilon_2, \tag{2}$$

*where $\epsilon$ is the logit gap after projection to the output embedding between $t$ and $t_*$. $\epsilon_1$ denotes the logit gap before edit and $\epsilon_2$ denotes the logit gap after edit. A value of $\epsilon \approx 2.30$ corresponds to a 90% top-1 prediction probability.*

**Remark 4.5.** *Lemma 4.4 suggests that an edited value should be first similar to the vector pointing from $t$ to $t_*$ after a projection with $W_V$. Then, the edited values $\Delta v$ should be sufficiently large to ensure that the success of the edit.*

Our assumption here simplifies the connection between the edited layer and the prediction layer, as in real-world scenarios, the edit layer might pass through subsequent layers and undergo multiple attention operations before finally connecting to the prediction layer. However, the path we're considering - from the edit layer to the prediction layer via an attention - is arguably the most direct route. We contend that this direct path is crucial and warrants particular attention and this simplification allows us to focus on the most immediate and potentially significant impact of edits.

**Lemma 4.6** (**Robustness Requirement for the Key-Values**). *If the newly added knowledge triplet $(h, r, t_*)$ can be robustly retrieved for all $k_s \in K_s$, it is required the following inequality to be satisfied:*

$$(w_{t_*} - w_t)^T W_V (k_s^T C^{-1} k_*) \cdot v_*^T > \epsilon_1 + \epsilon_2, \forall k_s \in K_s. \tag{3}$$

When we look into the lemma, $\alpha_{s,*} = k_s^T C^{-1} k_*$ can be seen as a similarity measure on a projected space. This lemma implies that (1) $v_*$ is decided by $\min_{k_{s'} \in K_s}(k_{s'} C^{-1} k_*)$, that is, $k_*$ should be near all $k_s \in K_s$. If not, $v_*$ needs to be of large magnitude to counter the difference. (2) $v_*$ needs not only to be aligned with the direction $(w_{t_*} - w_t)$, but also has a sufficiently large magnitude to ensure prediction success.

**Lemma 4.7** (Specificity Requirement for the Key-Values). *If the newly added knowledge triplet $(h, r, t_*)$ would not be retrieved for any $k_n \notin K_s$, it is required the following inequality to be satisfied:*

$$(w_n - w_{t_*})^T W_V (k_n^T C^{-1} k_*) \cdot v_*^T < \epsilon_3, \forall k_o \notin K_s \text{ and } \forall w \in W, \tag{4}$$

*where $t_n$ is the original target retrieved by $k_n$ and $\epsilon_3$ denotes the logit difference between $t_n$ and $t_*$.*

---

[1]In a real-world scenario, the edit cannot be lossless. Here, for a clear intuition, the above lemma is presented in an ideal way as the editing process will change the value of previously stored key-value pairs. We show that even considering the lossless scenario, the current LLMs cannot satisfy robustness and specificity requirements.

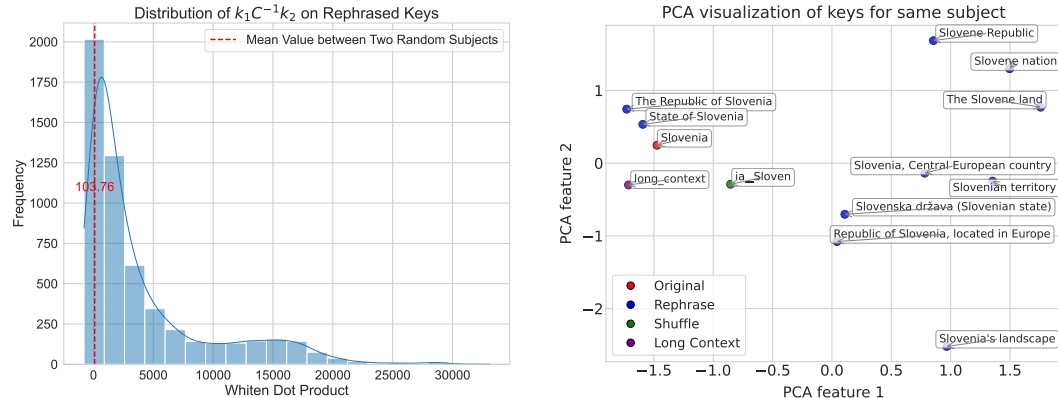

Figure 3: **Left:** The distribution of whitening similarity between two rephrases of a same subject. **Right:** A visualization for key representations of rephrases for 'Slovenia' in LLaMA2-7B.

One simple solution for this lemma is $k^T C^{-1} k_* = 0$, which describes no superposition, as discussed in one of the concurrent work (Hu et al., 2024a). However, as superposition generally exists among existing LLMs, we discuss more general cases here. The detailed proof of all above lemmas can be found in Appendix.

***Group Discussion.*** The aforementioned lemma can be understood through a simple conceptual model. Imagine each piece of knowledge as a group of individuals engaged in a discussion. Each person has an existing key-value pair in the model, where the key represents their position and the value represents the volume of their voice. When attempting to add new knowledge, it is necessary to introduce an individual at the appropriate position (key) to influence the discussion. To impact the conversation, this new individual must speak loudly —- both to counterbalance the voices of the others and to ensure that those positioned further away can hear them. This scenario can be further complicated by the presence of other nearby groups (representing distinct knowledge) that may be in close proximity to the original group. Consequently, the heightened vocal projection of the new participant has the potential to inadvertently influence the discussions occurring within these adjacent groups.

# 5 EMPIRICAL ANALYSIS: A BREAK OF REQUIREMENTS

In light of our theoretical results in previous section, we analyze the current knowledge editing methods, showing that the robustness and specificity requirements from previous section cannot be satisfied with inner representations as keys, motivating our approach.

## 5.1 EXPERIMENTAL SETUP

Following previous work, we use the CounterFact (Meng et al., 2023a) datasets, choosing LLaMA-2 as our base model. In addition to the prompt from CounterFact dataset, we additionally consider three types of perturbation in our experiments, namely prompt appended with unrelated long context, subject rephrase and random shuffled subject. Even though the shuffled subject does not contain the same semantic meaning, it demonstrates how keys shift when the position of same word occurs at different positions.

We collect 10 rephrases for each subject by prompting `gpt-4o-mini`. The prompt we use can be found in Appendix. For long context, we follow Ma et al. (2024c) and extract random text span of 512 tokens from Wikitext-103 Merity et al. (2016). For rephrased prompts, we use the paraphrases of prompts released by Patil et al. (2023). For shuffled subject, we sample 10 random orderings of tokens in the subject. We use 100 samples in our valid set for empirical analyses.

| Subject | Score | Prefix |
|---|---|---|
| *Michael Jordan* | 3898.8 | ... reach 10,000 career assists. Kobe |
| *Emmitt Smith* | 8469.9 | ... for the NL lead with Randy Johnson, Kevin Millwood, Tom Glavine |
| *Pasquale Di Sabatino* | 2190.6 | ... archbishop of Albi Giovanni Costanzio Caracciolo |
| *East China Normal University* | 6798.2 | ... Located southwest of Gongyi city in Gongxian County |
| *BMW Z3* | 3570.2 | ... her lead over Röhrl shrank to 18 minutes. The Toyota Celica |

Figure 4: **Left**: CounterFact subjects have unrelated prefixs which are close in keys. **Right**: Semantically similar subjects brings challenges to specificity.

## 5.2 Empirical Statistics of Keys, Values and Others

**Dissimilar Keys**  In Figure 3 (left), we present the distribution of whitening dot products $k_1 C^{-1} k_2, \forall k_1, k_2 \in K_s$. For each subject in CounterFact, we compute the dot product for each pair of keys of a subject's rephrases. We utilize the inputs to the FFN's down projection of layer 5 of LLaMA-2 as our keys, consistent with previous ROME experiments. Additionally, we include the dot product values of randomly sampled keys as a baseline for comparative analysis. Higher values indicate greater similarity between key pairs in the whitening space.

Analysis of the distribution reveals notable findings – While some key pairs exhibit high similarity, a substantial portion of the distribution mass concentrates near zero, aligning with the random keys baseline. This suggests a significant variability in the similarity of rephrased subject representations. Intriguingly, some key pairs demonstrate negative similarity. This phenomenon warrants further investigation to understand its implications for semantic representation and model behavior.

Figure 3 (right) provides a visualization of representations for the subject 'Delphine de Girardin' after three types of perturbations, reduced to two dimensions using Principal Component Analysis (PCA). This visualization corroborates our previous findings: (1) *Context sensitivity:* Long irrelevant context induces a slight shift in the representation, indicating contextual influence on subject encoding. (2) *Rephrase variability:* Rephrased versions of the subject sometimes cluster close to the original representation, while at other times they are distant. (3) *Order dependence:* Shuffling the word order results in substantial deviations from the original representation. This observation highlights the model's sensitivity to word order, even when the constituent tokens remain unchanged.

These findings challenges the intuition that semantically equivalent subjects should have similar representations, and poses severe challenges to the effectiveness of edits. When the edited key has near-zero or negative similarity with other keys, based on Lemma 4.6 it becomes virtually impossible for the edited value to be retrieved, potentially compromising the robustness of the edit.

**Similar but Non-Related Keys.**  We also investigate whether there exists different subjects that have highly similar keys. To this end, we iterate through a slice of Wikitext-103 dataset (~80M tokens) and select those are close to subjects in CounterFact in the whitening space. We filter those tokens whose prefix has the same subject token and collect the top-10 unrelated keys of each subject in CounterFact. The left of Figure 4 plots the distribution of whitening similarities between unrelated prefixs and CounterFact subjects. We find that a large portion of them has extremely high whitening similarity score, i.e., > 2500. Based on our theory, it indicates that any edit affects these subjects would inevitably affect the output on these unrelated prefixs.

On the right side of Figure 4, we show cases of subjects and their top-1 prefix in terms of whitening similarities. Interestingly, we observe that a subject can exhibit similarity in distributional semantics (Lenci & Sahlgren, 2023) to its corresponding top unrelated prefix. For example, the keys of *Michael Jordan* are highly similar to keys of a prefix related to *Kobe*. Considering that these two basketball players has much in common in many perspectives, it makes sense that their keys are similar. However, an edit to *Michael Jordan* affects *Kobe* would be definitely unreasonable.

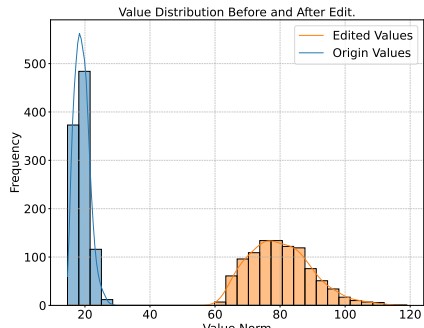

Figure 5: Values before and after edit with ROME.

**Loud Voices.** In Figure 5, we present the distribution of values before and after edits, using LLaMA-2 7B and ROME. The results demonstrate that post-edit values exhibit significantly larger L2 norms compared to pre-edit values. This observation aligns with our findings in Lemma 4.4 and 4.6, which suggest that edited values must be sufficiently large to effect changes on the current key and influence distant keys.

However, this increase in value magnitude, while necessary for effective editing, presents potential challenges. As indicated by Lemma 4.7 and our previous analysis, these 'loud' values may inadvertently affect unrelated keys, particularly those that are proximal in the representation space to the one being edited. This observation highlights a tension between achieving targeted edits and avoiding unintended consequences in the model's broader knowledge representation.

**Summary.** Our findings collectively suggest that the inner representations of large language models (LLMs) may not serve as reliable keys for editing purposes. The observed variability in key similarities, even among semantically equivalent subjects, coupled with the necessity for large-magnitude value changes, poses significant challenges for precise and controlled model editing. These issues can lead to unintended effects on unrelated parts of the model's knowledge and compromise the specificity of edits. Furthermore, the sensitivity of representations to word order and context underscores the instability of using these internal states as edit targets. These limitations motivate us to explore alternative approaches, particularly the concept of *branching a separate path for keys*. By creating a dedicated pathway for key representations, we may achieve more stable and controllable edit targets, potentially mitigating the issues of representation variability and unintended side effects observed when directly manipulating the model's inner representations.

## 6  ROBUST EDIT PATHWAY

Our idea to address above issues is to disentangle the keys from the models' inner representations, by introducing a potential branching path as keys of edited facts.

We achieve this by placing an adaptor directly after the keys and modify the representations of the keys when it is necessary. As shown in Figure 6, our adaptor consists of two modules, a *projection* module that is responsible for aligning the keys and a gate module that activates the adaptor when a token representation needs to be edit:

$$\hat{k} = f_{\text{gate}}(k) \circ f_{\text{proj}}(k) + k, \tag{5}$$

where $k \in \mathbb{R}^{bsz \times L \times D}$, $f_{\text{gate}}(k) \in \mathbb{R}^{bsz \times L \times 1}$ and $f_{\text{proj}}(k) \in \mathbb{R}^{bsz \times L \times D}$.

The gate mechanism here operates on the granularity of tokens and adaptively selects whether a key should be modified or not.

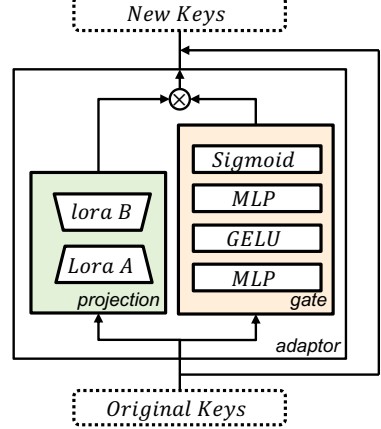

Figure 6: Architecture design of the adaptor.

We train the adaptor by aggregating the keys of same subject $k_s \in K_s$ toward our injected target key $k_*$:

$$\mathcal{L} = -|(\frac{\hat{k_s}}{||\hat{k_s}||_2})^T C^{-1} k_*| \tag{6}$$

where $\hat{k_s}$ is the output keys after adaptor. The intuition is inspired by Lemma 4.6 and 4.7. If the edited key is close to the keys of the same subejct, especially those we found dissimilar in Section 5.2, the edit would be more robust.

In practice, we find that model inclines to 'cheat' by simply increasin the norm of $k_s$ and thus we normalize the output of $f$. In practice, we take the last token of rephrased subjects over different

contexts and rephrased templates as $k_s$. This objective, built on the whitening similarity, further strengthen the validness of our theoretical results.

For testing, we use a gate threshold $\tau$ to determine whether to activate this branch. This gate mechanism allows the model to dynamically decide whether the original keys should be modified. If not, the keys are left intact and thus ensure the locality of edits.

Algorithm 1 illustrates the training and testing with our adaptor.

## 6.1 EXPERIMENTAL RESULTS

Table 1: REP improves over both in-domain and out-of-domain robustness metrics. $\tau = 0.9$

| | | Edit Performance | | In-Domain | | | Out-of-Domain | | |
|---|---|---|---|---|---|---|---|---|---|
| | | *Success* | *Locality* | *Rephrase* | *Shuffle* | *Long* | *Rephrase* | *Shuffle* | *Long* |
| LLaMA2-7B | ROME | 84.5 | 96.5 | 52.1 | 11.4 | 74.6 | 55.7 | 5.9 | 74.5 |
| | ROME + REP | 81.0 | 91.7 | **79.1** | **75.3** | **78.6** | **71.8** | **43.6** | **79.8** |
| Mistral-7B | ROME | 94.5 | 94.2 | 65.6 | 14.2 | 86.7 | 65.8 | 14.5 | 86.0 |
| | ROME + REP | 91.5 | 91.1 | **90.9** | **86.0** | **88.4** | **83.9** | **48.8** | **86.8** |

**Setup**    We evaluate our Robust Edit Pathway with two representative locate-and-edit methods, namely ROME and MEMIT. We use the LLaMA2-7B model as our base model and CounterFact as our dataset. We filter knowledge triplets of CounterFact not presented in the model as Meng et al. (2023a) did, and randomly sample 100 knowledge triplets as the validation set and 400 triplets as the test set. Our research concentrates on the single-edit paradigm, wherein one fact is injected into the LLM at a time. While other studies in model editing explore modifying multiple facts continuously (Mitchell et al., 2022c; Hartvigsen et al., 2023; Meng et al., 2023b), we have found that robustly injecting even a single fact presents significant challenges. Therefore, we maintain our focus on this fundamental aspect in this work.

For evaluation, we consider the following metrics: (1) *Success*: the ratio of targeted knowledge achieving the top probability; (2) *Locality*: the raito of related but non-identical facts kept intact by the edit; (3) *Rephrased*: the ratio of same subject with rephrased surface form achieving a success edit; (4) *Shuffled Subject*: the ratio of shuffled form of subjects achieving a success edit; (5) *Long Context*: the ratio of facts with random context appended before achieving a success edit.

Metrics (1) and (2) are widely used in previous studies (Meng et al., 2023a;b) of knowledge editing methods, where the success rate indicates the effectiveness and the locality represents the specificity of editing methods. Metrics (3)-(5) are robustness metrics aligned with our previous analyses. Improving these metrics suggests a more robust editing method. We report robustness metrics at both in-domain, where the test cases are seen in training adaptor, and out-of-domain, where the test cases are not seen by adaptor. Note that in our 'in-domain', we do not reveal the target knowledge to the model, we only aggregate the keys.

**ROME and MEMIT's Failure on Robustness.**    Our resutls are shown in Table 1. Our baseline methods, ROME, achieve solid edit success rate (~85%) and good locality scores (>90%) on both LLaMA2-7B and Mistral-7B. Nonetheless, these methods are easy to fail with robustness tests. ROME's success rate drops ~30% with rephrased subjects, drops ~70% with shuffled subjects ordering and drops ~10% with randomly appended long context. These results reconcile with those reported in previous studies (Ma et al., 2024c).

**Effectiveness of Robust Edit Pathway.**    Then, with a slight cost of *Success* and *Locality*, the robustness metrics siginificantly improve when adopting our method. Specifically, after aggregating the keys, the success rate of in-domain tests all reaches about 75-79%, which is close to the success rate without any perturbation (the upper bound). This further validates our theoretical and empirical results. Then, our method also improves over out-of-domain robustness tests, indicating that our adaptor learns to generalize across types of perturbations.

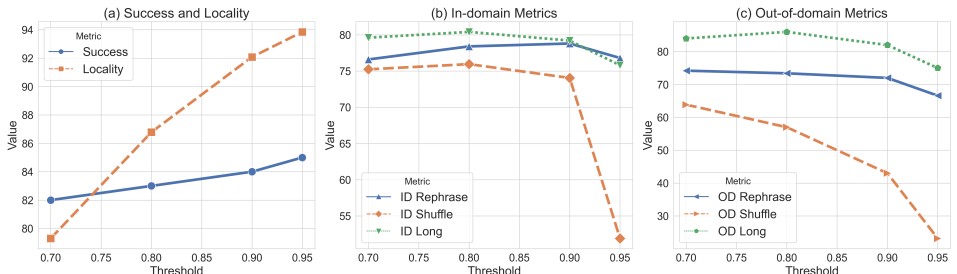

Figure 7: Validation performance of edited LLaMA2-7B against $\tau$ is plotted.

**Study of Gate Threshold.** Gate threshold $\tau$ is crucial to the performance of REP. Figure 7 varies different values of $\tau$ and evaluate our metrics. We find that a larger $\tau$ leads to a better locality and success rate. Meanwhile, the robustness metrics first plateau then degrade with the increase of $\tau$, indicating a trade-off between robustness and edit performance.

# 7  CONCLUSION AND FUTURE DIRECTIONS

In this work, we have presented a comprehensive analysis of the challenges faced by knowledge editing techniques in large language models, specifically focusing on the robustness issues in locate-and-edit methods. Our theoretical framework, centered on the 'group discussion' model, provides valuable insights into the mechanisms of knowledge insertion and the requirements for robust and specific edits.

The proposed Robust Edit Pathway (REP) addresses the key limitations identified in our empirical analysis, effectively disentangling editing keys from the LLM's internal representations. Our extensive evaluations on LLaMA2-7B and Mistral-7B demonstrate significant improvements in robustness across various metrics, including rephrased subjects, shuffled token ordering, and long context additions, while maintaining high success rates and locality.

These results not only validate our theoretical foundations but also offer a promising direction for enhancing the reliability and flexibility of knowledge editing in LLMs. The REP's ability to generalize across different types of perturbations represents a substantial step forward in addressing the critical robustness issues prevalent in existing techniques.

Future work could explore the scalability of our approach to larger models and more diverse datasets, as well as investigate potential applications in continual edits and dynamic knowledge updating for LLMs. Overall, this research contributes to the ongoing efforts to create more adaptable and trustworthy language models, paving the way for their responsible deployment in real-world applications.

# ETHIC STATEMENT

We honor the ICLR Code of Ethics. No private data or non-public information was used in this work.

# REPRODUCTION STATEMENT

We have appended the data and code in supplementary files for review and reproduction.

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

# A PROOFS

## A.1 PROOF OF LEMMA 4.1

Since $K$ has full row rank (rank$(K) = D_1$), $KK^T$ is invertible. To find $\alpha$, we use the Moore-Penrose pseudoinverse of $K$.

Given $K \in \mathbb{R}^{D_1 \times N}$, the pseudoinverse $K^+$ is defined as: $K^+ = K^T(KK^T)^{-1}$, which also minimizes $||K\alpha - \hat{k}||$.

Then, we can express $\hat{v}$ as:

$$\hat{v} = V\alpha = VK^T(KK^T)^{-1}\hat{k}. \tag{7}$$

Note that since $N \gg D_1$, the system $K\alpha = \hat{k}$ is underdetermined. This means there are infinitely many solutions for $\alpha$, and the Moore-Penrose pseudoinverse gives the one with the smallest norm.

**Algorithm 1** Branching Adaptor

1: **function** GATE($k_s$)                                          // Compute gate module output $s$.
2:      $o = \text{Linear}(\text{GELU}(\text{Linear}(k_s, \theta_{m2})), \theta_{m1})$
3:      $s = \text{Sigmoid}(o)$
4:      **return** $s$
5: **end function**
6: **function** PROJECTION($k_s$)                                    // Compute low-rank projection output $x$.
7:      $x = \text{Dropout}(k_s)$
8:      $x = \text{Linear}(\text{Linear}(x, \theta_{p1}), \theta_{p2})$
9:      **return** $x$
10: **end function**
11: **function** TRAIN_BPA($k_s$)
12:      Initialize $[\theta_{p1}, \theta_{p2}, \theta_{m1}, \theta_{m2}]$
13:      **for** steps = 1 to $N$ **do**
14:          $\hat{k_s} = \text{PROJECTION}(k_s) \circ \text{GATE}(k_s)$
15:          Maximize: $\mathcal{L} = -|(\frac{\hat{k_s}}{||\hat{k_s}||_2})^T C^{-1} k_*|$        // Aggregate keys toward the injected $k_*$.
16:      **end for**
17:      **return** $[\theta_{p1}, \theta_{p2}, \theta_{m1}, \theta_{m2}]$
18: **end function**
19: **function** TEST_BPA($k_s$)
20:      $x = \text{PROJECTION}(k_s)$
21:      $s = \text{GATE}(k_s)$
22:      $m = \text{int}(o \geq \tau)$                                // Testing with binary gate.
23:      **return** $x \circ m$
24: **end function**

### A.2  PROOF OF LEMMA 4.4

We can focus on the logit difference between the largest and the second-largest logits to achieve high confidence in the final prediction. This difference is an important factor in determining the confidence of a prediction in a softmax layer.

Here, we simplify the modeling by only considering the contribution of edited layer towards final prediction via its the edited layer is connected to the final prediction layer directly via its attention layer

Given a vector of logits $\mathbf{z} = [z_1, z_2, \ldots, z_n]$, the softmax function yields probabilities $\mathbf{p} = [p_1, p_2, \ldots, p_n]$, where:

$$p_i = \frac{e^{z_i}}{\sum_{j=1}^n e^{z_j}}$$

To increase the confidence in the prediction for the largest logit, maximize the difference between the largest logit and the second-largest logit.

Let $z_{\max}$ be the largest logit and $z_{\text{other}}$ be another logit. The logit difference $\Delta$ is given by: $\epsilon = z_{\max} - z_{\text{other}}$.

The softmax confidence for the class corresponding to $z_{\max}$ can be expressed as:

$$p_{\max} = \frac{e^{z_{\max}}}{e^{z_{\max}} + e^{z_{\text{other}}} + \sum_{k \neq \max, \text{ other}} e^{z_k}} \tag{8}$$

$$< \frac{e^{z_{\max}}}{e^{z_{\max}} + e^{z_{\text{other}}}} \tag{9}$$

$$= \frac{1}{1 + e^{-\epsilon}} \tag{10}$$

After organizing between two sides, we get a lower bound of $\epsilon$ for achieving a sufficiently large confidence:

$$\epsilon > -\log(1 - \frac{1}{p_{\max}}) \tag{11}$$

Now, in a transformer architecture, the edited MLP layer is connected to the word prediction layer through an attention layer at the final token. Let the difference between the original and the edited output of the MLP layer be $\Delta v$, the parameters of the attention layer are $W_Q, W_K, W_V \in \mathbb{R}^{D \times D}$ and the query vector at the prediction token is $q = Qh$, the attention layer's output is defined by

$$o = \sum_j \text{Softmax}(q^T W_K v_j) W_V v_j. \tag{12}$$

Since in the locating part we use causal intervention to identify the most influential position of tokens to edit, we can assume that $(q^T W_K v_s)$ has already get the largest weight. The difference caused by edited MLP is,

$$\Delta o = \text{Softmax}(\cdot) W_V \Delta v. \tag{13}$$

Then, residual connections directly connect this output to the final word prediction layer. Combining our result from equation 11, let the original fact $t$ before the edit has a logit gap $\epsilon_1$ and the new fact $t_*$ after edit has $\epsilon_2$, we can bound the $\Delta o$ with,

$$\begin{cases} (w_t - w_{t_*})^T o_{ori} > \epsilon_1 \\ (w_{t_*} - w_t)^T (o_{ori} + \Delta o) > \epsilon_2 \end{cases} \tag{14}$$

$$\Rightarrow (w_{t_*} - w_t)^T \Delta o > \epsilon_1 + \epsilon_2 \tag{15}$$

$$\Rightarrow (w_{t_*} - w_t)^T \text{Softmax}(\cdot) W_V \Delta v > \epsilon_1 + \epsilon_2 \tag{16}$$

$$\Rightarrow (w_{t_*} - w_t)^T W_V \Delta v > \epsilon_1 + \epsilon_2 \tag{17}$$

$$\tag{18}$$

Given that the softmax weight is at most 1, we have our lower bound on $\Delta v$.

### A.3 Proof of Lemma 4.6 and 4.7

Based on our result above, we can further derive results for

## B Experimental Details

### B.1 Data Construction

We build our evaluation data based on the CounterFact dataset. We further augment our data with all three robustness tests. For rephrased subjects by prompting `gpt4o-mini` with the following prompt.

> Give 10 rephrases representing the same entity: {ENTITY}

The unrelevant long contexts are extracted from the Wikitext-103 dataset (Merity et al., 2016). The shuffled tokens are generated via sampling different word ordering. Finally, we filter the samples that are not present in the current LLM, that is, given the prefix, the target tokens are not predicted by the LLMs with the top-1 probabilities. We sample 100 samples for validation and 400 samples for test. To evaluate in-domain and out-of-domain robustness, we split the all three kinds of robustness queries in a 50-50 manner. For each sample, we have 5 in-domain queries and 5 out-of-domain queries.

### B.2 Details of Training BPA

We implement our methods based on EasyEdit (Wang et al., 2023). We use Adam optimizer for all experiments and the learning rate is 5e-4. We train each adaptor for 10 steps. The inner dimension of the projection module is 32, and the inner dimension of gate module is 0.1 of key dimension.

