# OpenReview forum: "Keys to Robust Edits: From Theoretical Insights to Practical Advances"
_ICLR.cc/2025/Conference — ICLR 2025 Conference Withdrawn Submission_

### Official Review · Reviewer_8nJm · 2024-10-30

**Soundness:** 1
**Presentation:** 1
**Contribution:** 2
**Rating:** 3
**Confidence:** 4

**Summary:**

In this paper, the authors tackle the problem of knowledge editing in an LLM, i.e., how to replace one knowledge with another by modifying the weights. They propose a revision of the ROME approach based on theoretical results. Their approach consists of adding a key adaptor to the network to fix current problems with ROME.
Studying knowledge editing is thrilling, and theoretical analysis is the correct way to find valuable insights and improvements. However, the paper needs to be more mature, with many imprecisions, presentation issues, typos, and missing information.

**Strengths:**

S1. The authors provided the code for their approach. However, they could have written a README to understand the content better. Besides, the code is not anonymized, with paths containing the programmer's name (hardcoded paths are not a good sign of reproducibility).

S2. The subject is well-motivated at the beginning.

S3. In several places, the authors try to link the practical and the theoretical side of their work.

**Weaknesses:**

W1. The authors need to improve the presentation and clarity of the paper. There are many typos in the paper (see below), and some are in the theoretical results, which is concerning. Sections 3 and 4 take a lot of work to follow. For Section 3, the authors copy-pasted the content of ROME. However, it is tough to understand without the context of the ROME paper. Section 4 is a succession of Lemmas and corollaries barely connected. The authors need to explain the intuition better. I provide additional issues below.

W2. The experiment section is slim, with few baselines compared. The authors wanted to include MEMIT in the comparison, but it appears only in the text rather than in the tables. It is hard to understand what happened here. Also, a better benchmark for this task now exists: the KnowEdit dataset.

W3. The empirical analysis lacks quantitative evidence. Some conclusions are only drawn from one example.

W4. Many notions are used without being properly introduced (for example, the whitening similarity).

Typos/Others

T1. Be careful with apostrophes (often, a closing one is used instead of an opening one).

O2. Maybe write a summary of the contributions at the end of the introduction.

T2. L130: utlize -> utilize

T3. L132: adaoptor -> adaptor

T4. In definition 3.1, line 164, k should be f.

T5. In Remark 3.3, the sum is over j, not i.

O6. Figure 2 is never mentioned in the text.

O7. For Lemma 4.1, the proof is confusing. Simply say that WK=V, so W = VK^+. Then, \hat{v} = W \hat{k} = VK^+ \hat{k}.

O8. Lemma 4.1 introduced W, but it has not been used.

T9. Lemma 4.1: The sum starts at 0, not i.

O10. At Corollary 4.2, it is hard to get what K_s and V_s are and what the "same semantic meaning" means.

O11. In Corollary 4.2, is alpha computed using K_s and V_s, or just K and V. If it is K_S and V_s, the assumption N>>D might not hold,
particularly in the experiments.

O12. In Lemma 4.4, epsilon is not used. I do not see why we give a value to epsilon.

O13. In Lemma 4.4, is the assumption reasonable?

O14. Equation 11 cannot be correct as 1 - 1/p <= 0. Does it have an impact on what follows?

O15. Line 773, I need help understanding how Equation 11 is used and where epsilon is.

O16. In Lemma 4.6, what does "robustly retrieved" mean?

O17. In the Appendix, the authors did not finish the proofs (Section A3). The sentence ends in the middle, which is not serious.

O18. In Figures 3 and 4, on the left, I think the authors should normalize the results as it is usually done with similarities.

O19. Figure 3, on the left, is hard to read. In black and white, it is impossible. Even in the color version, some dots are not visible.

O20. What is the whitening similarity? Where is it defined?

O21. Line 342, how is K_s defined?

O22. Line 348, where is the baseline? It does not appear in the plots.

O23. Line 352, the example in the text (Delphine de Girardin) does not match the example in Figure 3 (Slovenia).

O24. I don't think PCA is relevant here for analyzing a single subject. By default, PCA will spread the points. The distances are only meaningful relative to other distances. The authors should perform PCA on several subjects before being able to analyze the results. Also, PCA can modify what happens in higher dimensions.

O25. Did the authors perform a quantitative analysis for Section 5.2? One cannot conclude from a single example. There should be average distances and metrics.

T26. There are some random ~ over numbers.

T27. Line 367, remove are.

O28. Line 368, "...prefix has the same subject token" as what? Give more examples here.

O29. Line 370, "high whitening noise." Compared to what?

O30. On which dataset?

O32. Line 419, what are bsz, L, and D?

T33. Line 430, "g" missing.

T34. Line 433, "theorectical" -> theoretical

O33. Line 435,  "whether to activate this branch". Are we talking about projection only?

O34. Table 1, why are the results in the first column not in bold? It is ok not always to have the best results.

O35. Are the results in Table 1 statistically significant? Can we have a standard deviation?

O36. Section 6.1, where are the results for MEMIT?

O37. Line 453. "We use LLaMa2" and Mistral!

O38. How big is the train set?

O39. Why did not the authors use all the metrics in the ROME paper?

T40. Line 432, raito -> ratio. Line 474, resutls -> results (just use an autocorrector).

O41. The authors should add more baselines to the experiments.

O42. The authors should include an error analysis section. Where are the errors coming from? For example, does the rephrasing always make sense? Does the shuffling always make sense? What would a human say?

O43. For the references, try to use the real reference and not only the Arxiv link.

**Questions:**

See above.

---

### Official Review · Reviewer_GtW6 · 2024-11-04

**Soundness:** 2
**Presentation:** 2
**Contribution:** 3
**Rating:** 5
**Confidence:** 3

**Summary:**

Analysis of knowledge editing techniques in LLMs.
Robustness issues in locate-and-edit models.
Propose Robust Edit Pathway addresses those issues.

**Strengths:**

Knowledge editing is important for improving the quality of LLMs
Related work seems well covered and reviewed
The solution proposed sheds light on the key-value approaches
The paper is largely written, or heavily edited, by a LLM

**Weaknesses:**

The nature of key-value data model would require an assessment of the conceptual modelling issues involved in knowledge editing
A lot of knowledge is not stored in individual triple, but in subgraphs
English issues and instability of terminology/definitions
Few, poor examples do not help the reader, e.g., the triple (USA, president-of, Biden) is counterintuitive unless we interpret "president-of" a "has-president"

**Questions:**

What kind of knowledge do you intend to edit?
What dynamics "group discussion" would cover?

---

### Official Review · Reviewer_d9VQ · 2024-11-08

**Soundness:** 3
**Presentation:** 3
**Contribution:** 3
**Rating:** 6
**Confidence:** 3

**Summary:**

This paper works on the robustness challenge in knowledge editing within large language models, particularly using locate-then-edit methods.
The authors propose a theoretical framework and propose a "Robust Edit Pathway" (REP). It separates editing keys from LLM representations, allowing more robust and specific edits.
Experimental results demonstrate that REP improves robustness metrics on Llama2-7B and Mistral-7B models with the CounterFact dataset.
It also achieves higher performance in rephrased and long-context scenarios.

**Strengths:**

1. The proposed REP framework uses a project and gate mechanism and separates editing pathways, providing a new approach to the knowledge editing task.
2. The paper combines theoretical derivation and empirical evidence.
3. The paper reports experiments and shows the proposed REP works well on a representative locate-then-edit method ROME.

**Weaknesses:**

1. The experiments mainly use Llama-2-7B and Mistral-7B. Maybe more LLMs can be included and tested.
2. The experiments solely use the CounterFact dataset. We expect to see the results of real-world knowledge editing. This is because knowledge editing in the real world tends to be more diverse and complex.
3. What are the failure cases of REP? Including failure cases of editing and robustness tests.
4. It seems the experiment section only reports results with ROME and ignores MEMIT.

**Questions:**

1. Table 1 only includes ROME. Where are the results of MEMIT?
2. The paper only uses the CounterFact dataset for experiments. How about the performance of real-world facts like [here](https://arxiv.org/pdf/2402.18909)?
3. In the Figure 4 caption, "prefixs" should be "prefixes".
4. Line 431, "increasin" should be "increasing".
5. Line 799, "unrelevant" should be "irrelevant".

---

### Author Response · Authors · 2024-11-21

We are greatly thankful for all the reviewers' insightful comments. They help us a lot.
We decided to withdraw our paper from ICLR to polish it further.

Authors

---

### Note · Authors · 2024-11-21

I have read and agree with the venue's withdrawal policy on behalf of myself and my co-authors.